# Photocatalytic Degradation of Diclofenac Using Al_2_O_3_-Nd_2_O_3_ Binary Oxides Prepared by the Sol-Gel Method

**DOI:** 10.3390/ma13061345

**Published:** 2020-03-16

**Authors:** José Eduardo Casillas, Jorge Campa-Molina, Francisco Tzompantzi, Gregorio Guadalupe Carbajal Arízaga, Alejandro López-Gaona, Sandra Ulloa-Godínez, Mario Eduardo Cano, Arturo Barrera

**Affiliations:** 1Departamento de Ciencias Básicas, Centro Universitario de la Ciénega, Universidad de Guadalajara, Av. Universidad, No. 1115, Ocotlán C.P. 47820, Jalisco, Mexico; duartcasillas88@gmail.com (J.E.C.); meduardo2001@hotmail.com (M.E.C.); 2Departamento de Electrónica, Universidad de Guadalajara, Marcelino García Barragán 1422, Guadalajara C.P. 44430, Jalisco, Mexico; jcampamolina@gmail.com (J.C.-M.); sago2063@gmail.com (S.U.-G.); 3Departamento de Química, Universidad Autónoma Metropolitana—Iztapalapa, San Rafael Atlixco 189, Ciudad de México C.P. 09340, Mexico; fjtz@xanum.uam.mx (F.T.); gaby@xanum.uam.mx (A.L.-G.); 4Departamento de Química, Universidad de Guadalajara, Marcelino García Barragán 1422, Guadalajara C.P. 44430, Jalisco, Mexico; gregoriocarbajal@yahoo.com.mx

**Keywords:** Al_2_O_3_-Nd_2_O_3_, sol-gel, diclofenac, photodegradation, structure, band-gap energy

## Abstract

This paper reports the sol-gel synthesis of Al_2_O_3_-Nd_2_O_3_ (Al-Nd-*x*; *x* = 5%, 10%, 15% and 25% of Nd_2_O_3_) binary oxides and the photodegradation of diclofenac activated by UV light. Al-Nd-based catalysts were analyzed by N_2_ physisorption, XRD, TEM, SEM, UV-Vis and PL spectroscopies. The inclusion of Nd_2_O_3_ in the aluminum oxide matrix in the 10–25% range reduced the band gap energies from 3.35 eV for the γ-Al_2_O_3_ to values as low as 3.13–3.20 eV, which are typical of semiconductor materials absorbing in the UV region. γ-Al_2_O_3_ and Al-Nd-x binary oxides reached more than 92.0% of photoconverted diclofenac after 40 min of reaction. However, the photocatalytic activity in the diclofenac degradation using Al-Nd-*x* with Nd_2_O_3_ contents in the range 10–25% was improved with respect to that of γ-Al_2_O_3_ at short reaction times. The diclofenac photoconversion using γ-Al_2_O_3_ was 63.0% at 10 min of UV light exposure, whereas Al-Nd-15 binary oxide reached 82.0% at this reaction time. The rate constants determined from the kinetic experiments revealed that the highest activities in the aqueous medium were reached with the catalysts with 15% and 25% of Nd_2_O_3,_ and these compounds presented the lowest band gap energies. The experimental results also demonstrated that Nd_2_O_3_ acts as a separator of charges favoring the decrease in the recombination rate of electron-hole pairs.

## 1. Introduction

Aluminum oxide is an insulating material with band gap energy values above 5 eV [1,2,3]. The energy in this gap can be reduced to values as low as 2.5 eV by adjusting the content of defects on its surface [4,5,6]. According to previous reports, the modified surface can be considered a new phase with different properties associated to new surface chemistry and chemical activity [4,5,6]. We have reported that the sol-gel method is suitable to produce γ-Al_2_O_3_ with photocatalytic activity to degrade organic pollutants in water [7,8,9], however it is desirable to enhance photocatalytic performance in order to reach a higher conversion in the lowest time. In earlier works, the photocatalytic activity of γ-Al_2_O_3_ in the degradation of phenolic compounds has been improved by impregnating noble metal (PdO or Ag^0^) onto Al_2_O_3_-Ln_2_O_3_ (Ln = Nd or Gd) binary oxides [8,10,11] or by doping Al_2_O_3_-Nd_2_O_3_ with the semiconducting ZnO to form Al_2_O_3_-Ln_2_O_3_-ZnO ternary oxides [9]. Although the experimental evidences demonstrated that the photocatalytic activity of γ-Al_2_O_3_ enhances with the addition of rare earth oxides, semiconducting oxides, or noble metals, the performance against degradation of drug pollutants in water is still unknown; therefore, we proposed to study the photocatalytic degradation of diclofenac in aqueous medium catalyzed with Al_2_O_3_-Nd_2_O_3_ binary oxides and the efficiency related to the Nd_2_O_3_ content.

Diclofenac is an anti-inflammatory drug used frequently by human beings which is discharged into the wastewater causing further pollution in different sources of drinking water [12,13,14], since it is very stable against the conventional treatments of water [15,16]. The persistence in ecosystems occurs due to the low biodegradability and this is also a reason for the incomplete removal in water treatments based on microbiological activity [14,15,16,17]. Therefore, more efficient methods to eliminate this molecule are needed, and the photodegradation arises as an alternative. Then, this work studies the preparation of aluminum oxides with different contents of Nd_2_O_3_ and the relationship with the photocatalytic activity to degrade diclofenac in aqueous solutions. The characterization of the binary oxides by different solid-state techniques will be presented.

## 2. Materials and Methods

### 2.1. Synthesis

The sol-gel method was selected since this procedure allows for the simultaneous formation of metal oxides [8,9,10,11]. First, a solution with a certain amount of aluminum tri-sec-butoxide (97.0%, Aldrich, St. Louis, MO, USA) and 10 mL of 2-methylpentane 2,4-diol (99.999%, JT Baker, Phillipsburg, USA) as a complexing agent was stirred in a round glass flask for 60 min while the temperature was kept at 70 °C. Thereafter, the temperature was reduced to 50 °C and mixed with a second solution prepared by dissolving neodymium acetylacetonate (99.99%, Aldrich) in toluene (99.999%, Aldrich) at 40 °C. The resulting solution was stirred at 50 °C, then, after 60 min, 10 mL of deionized water was added dropwise to promote the hydrolysis of the aluminum and neodymium reagents to produce a gel. This gel was aged by heating first at 55 °C for 2 h and then at 80 °C for 12 h. The product at this step was a powder, which was dried 110 °C for 12 h and then calcined under air atmosphere at 650 °C for 4 h. The Al_2_O_3_-Nd_2_O_3_ binary oxides obtained after this process were labeled as Al-Nd-*x*, where x represents the mass percentage of Nd_2_O_3_ in the Al_2_O_3_ matrix. The content of Nd_2_O_3_ was 5%, 10%, 15% and 25%.

Bare Al_2_O_3_ (labeled as Al) was prepared by the same sol-gel procedure with aluminum tri-sec-butoxide, whereas the Nd_2_O_3_ reference was obtained by calcining neodymium acetylacetonate at 800 °C for 5 h.

### 2.2. Solid State Characterization

Textural properties of the photocatalysts were studied by nitrogen physisorption at the saturation temperature of liquid nitrogen (−195.6 °C) with Quantachrome Autosorb equipment, model IQ. The samples were conditions by degassing with helium at 300 °C for 5 h.

The structural property oxides were studied by powder X-ray diffraction with an STOE diffractometer, model Theta-Theta operated with Cu Kα (λ = 0.154 nm) radiation. The analysis was conducted in the 2θ mode with a 2θ step of 0.02° and collecting time of 60 s. High-resolution transmission electron microscopy (HRTEM) micrographs were acquired with a HRTEM FEI TECNAI F30 STWIN G2 microscope (Hillsboro, USA) with a power of 300 kV and a resolution of 24 nm. The morphology was studied with a scanning electron microscope (SEM) using a HRSEM Jeol 7600F microscope (Tokyo, Japan) operated with 30 kV. The chemical composition was determined with an energy dispersive X-ray spectroscopy (EDS) detector coupled to the microscope.

The absorption of UV-vis light necessary to determine the optical properties of the photocatalyst was measured with an UV-Vis CARY 300 (Varian, Victoria, Australia) spectrophotometer. The scanning was conducted with a speed of 600 nm per minute and resolution of 1 nm. The spectra were treated with the Kubelka-Munk function to determine the absorption edge and absorbance according to Equation (1), where E_g_ is the band gap energy, α is the measured absorbance, the product *hv* corresponds to the photon energy, and B is a constant for these materials [18,19]:α*hv =* B(*hv* − E_g_)^2^(1)

From this equation the (α*h*ν)^1/2^ and *h*ν values are plotted, and the linear part is extrapolated to determine the band gap energy. This relationship is known as the Tauc graph [20].

The photoluminescence spectra were obtained with a Fluorescence Varian Cary eclipse spectrophotometer using an excitation wavelength of 270 nm.

### 2.3. Photocatalytic Activity

The degradation of diclofenac was selected as the model to study the photocatalytic activity of the oxides, and the reactions were conducted in round bottom cylindrical glass with 1 L of capacity assembled to a glass jacket with double wall. The photodegradation of diclofenac (99.9%, Sigma-Aldrich) was studied with 80 ppm aqueous solutions. Before the reactions, this solution was bubbled with air during 12 h, and then 200 mL was transferred to the cylindrical glass and mixed with 200 mg of the Al-Nd-*x* sample representing a concentration of 1 mg of catalyst mL^−1^. The diclofenac solution was stirred and air was injected at a rate of 1 mL s^−1^, then a UV lamp (UVP products) emitting a 254 nm light with irradiance of 4400 μW cm^−2^ was introduced into the diclofenac solution. The photodegradation was followed as a function of time; for this, aliquots were removed every 5 min using a filter for the first 20 min, thereafter, the aliquots were collected at intervals of 10 min for a period of 60 min. The samples were analyzed by UV-Vis spectroscopy in a CARY 300 spectrophotometer. The quantification of diclofenac was based on the absorption at 275 nm and the kinetic was studied for a total period of 80 min of irradiation; γ-Al_2_O_3_ and Nd_2_O_3_ were used as references. As commonly reported for photodegradation of organic compounds in aqueous medium [21,22], the data were treated with the pseudo first-order equation of the Langmuir-Hinshelwood model:**Ln (C_0_/C) = k_app_t**(2)

## 3. Results and Discussion

### 3.1. Textural Properties of Photocatalysts

The textural properties of pure γ-Al_2_O_3_ and Al-Nd-*x* binary oxides are shown in Figure 1. Bare γ-Al_2_O_3_ shows a BET specific surface area (S_BET_) of 250 m^2^ g^−1^ (Figure 1A). The S_BET_ of γ-Al_2_O_3_ increased with neodymium oxide concentration to reach a maximum S_BET_ (266.0 m^2^ g^−1^) in the Al-Nd-10 sample. The increment of the S_BET_ value indicates that neodymium oxide enhances the textural features of γ-Al_2_O_3_ at low Nd_2_O_3_ concentration (≤10%). A drop in the S_BET_ was observed at higher Nd_2_O_3_ concentration (>10%), reaching a S_BET_ = 205.0 m^2^ g^−1^ in the Al-Nd-25 binary oxide. The pronounced drop in the S_BET_ of binary oxides at high Nd_2_O_3_ concentration might be due to the blocking of the alumina porous structure by the presence of Nd_2_O_3_ nanoparticles that are highly dispersed among the alumina agglomerates. The specific mesopore volume of γ-Al_2_O_3_ (Vmeso = 1.1 cm^3^ g^−1^) decreased with the Nd_2_O_3_ concentration reaching a Vmeso = 0.5 cm^3^ g^−1^ in the Al-Nd-25 binary oxide (Figure 1B), whereas the specific micropore volume was approximately constant (Vmicro = 0.3 cm^3^ g^−1^) for all the materials. The average pore size (APS) of Al-Nd-*x* binary oxide also decreased with neodymium oxide concentration from 12.0 nm in the bare γ-Al_2_O_3_ to 6.0 nm in the Al-Nd-25 sample (Figure 1C).

### 3.2. Crystalline Structure of Catalysts

Figure 2 shows the XRD patterns of the Al-Nd-*x* binary oxides and the references Al_2_O_3_ and Nd_2_O_3_. The aluminum oxide phase corresponded to γ-Al_2_O_3_ identified with the pattern recorded in the JCPDS card 10-0425, whilst the neodymium oxide sample contained the A-type hexagonal structure with cell parameters a = 3.827 Å and c = 5.991 Å (JCPDS card: 75-2255) [23]. Regarding the binary oxides, the patterns presented diffraction peaks at 2θ angles between 31.9° and 66.4° corresponding only to the γ-Al_2_O_3_ phase (JCPDS card: 10-0425). This observation suggests that small Nd_2_O_3_ nanoparticles are highly dispersed among the γ-Al_2_O_3_ agglomerates, which are below the limit of detection by XRD. Additionally, the reflections associated to the γ-Al_2_O_3_ phase decrease as the concentration of Nd_2_O_3_ increases, indicating a reduction of the crystalline quality of the alumina structure. It seems that a high Nd_2_O_3_ concentration (>10 wt%) in the Al-Nd-*x* binary oxide causes a disorder in the γ-Al_2_O_3_ structure, leading to a more disordered material.

The HRTEM micrograph was useful to identify the presence of the Nd_2_O_3_ phase with the A-type hexagonal structure. Figure 3A presents clear crystalline regions with defined grain boundaries. Additionally, the square inset in the same figure contains fringes of diffraction points; therefore, the existence of well-crystallized Nd_2_O_3_ is confirmed. The interplanar distance (d*_hkl_* = 0.32 nm) identified in the inset corresponds to the crystalline plane *hkl*(100) of Nd_2_O_3_ of the A-type hexagonal structure (24-0779 JCPDS). Furthermore, the HRTEM micrograph of the Al-Nd-15 sample (Figure 3B) corroborates the presence of nano-crystalline domains of Nd_2_O_3_ denoted by dark regions of approximately 8.0–10.0 nm in size, which are well mixed with the alumina agglomerates forming a highly disordered material at this high Nd_2_O_3_ concentration.

The high dispersion of small Nd_2_O_3_ nanoparticles among the γ-Al_2_O_3_ agglomerates is confirmed by the EDS image of the Al-Nd-15 binary oxide (Figure 4A) and by the mapping of the element Nd Mα of the sample (Figure 4B) exhibiting a high dispersion of Nd_2_O_3_ nanoparticles denoted by tiny spots in color blue over big γ-Al_2_O_3_ agglomerates of approximately 200 µm in size. It seems that 15 wt.% of Nd_2_O_3_ is the limit concentration in order to obtain a high dispersion of small Nd_2_O_3_ particles among the alumina agglomerates, although a disordered material is formed at high Nd_2_O_3_ concentration (>15 wt.%). This critical concentration was also observed with Al_2_O_3_-La_2_O_3_ materials prepared by the sol-gel method [24]. 

In order to get an insight into the morphology of the materials, SEM images of the γ-Al_2_O_3_, Al-Nd-15 and Al-Nd-25 binary oxides and Nd_2_O_3_ are shown in Figure 5A–D. The SEM micrograph of γ-Al_2_O_3_ exhibits a highly fibrous homogeneous material (Figure 5A). Meanwhile, the SEM micrograph of the Al-Nd-15 binary oxide shows a disordered material composed of alumina granular agglomerates of different sizes (Figure 5B). The morphology of the material changes at higher Nd_2_O_3_ concentration and the SEM micrograph of Al-Nd-25 binary oxide (Figure 5C) displays the formation of pieces of calcareous hollow material of different sizes, while the SEM micrograph of bare Nd_2_O_3_ depicts a highly crystalline material composed of plane layer agglomerates (Figure 5D).

A SEM mapping by element throughout an egg shell-type hollow spheroidal particle of the Al-Nd-25 binary oxide of Figure 5C indicates that aluminum and oxygen are compactly and evenly distributed, keeping the morphology of the particle (Figure 6), whereas neodymium is highly dispersed over the whole of the particle and carbon is densely populated throughout and outside of the particle. It is inferred that a high Nd_2_O_3_ concentration in the binary oxide, the calcination temperature, and the residual carbon from the 2,4-pentaneodiol used as a chemical modifier during the synthesis of the materials by the sol-gel method might result in different morphologies of the particles, in particular the residual carbon from the 2,4-pentaneodiol might act as a template giving rise to the framework of the hollow spheroidal particles.

### 3.3. Band Gap Energy of Al_2_O_3_-Nd_2_O_3_ Binary Oxides

The UV-Vis spectra of the γ-Al_2_O_3_ and Nd_2_O_3_ references as well as the Al-Nd-*x* binary oxides are shown in Figure 7. The spectrum of γ-Al_2_O_3_ presented a broad absorption in the 240–330 nm range, which is associated with electronic charge transfer and also with the accumulation of defects in the structure of this compound [25]. Regarding the Nd_2_O_3_ sample, the spectrum contains an intense band in the 200–290 nm range associated with an exchange of electrons between the valence and conduction bands [9,26]; also, a set of sharp absorption bands between 330 and 760 nm correspond to electron transitions within the 4f shell of neodymium [3,27]. 

Regarding the spectra of the binary oxides, the band between 240 and 315 nm from the alumina is present. In contrast, the intense and narrow absorption bands between 320 and 740 nm due to internal transitions in Nd^3+^ are slightly shifted to lower wavelengths due to a chemical interaction between Nd_2_O_3_ and γ-Al_2_O_3_ phases [9,27,28]. The absorption band in the UV region of pristine Nd_2_O_3_ was used to determine the band gap energy (E_g_) in the mixed oxides as these signals are produced by transference of electrons between the valence and conduction bands [9,26,29] and because this band is clearly defined in all the spectra and it is overlapped with the absorption band of γ-Al_2_O_3_. The plots to determine the E_g_ according to the Tauc equation [20] are depicted in Appendix A, whereas the E_g_ values are shown in Table 1.

The band gap energy value of γ-Al_2_O_3_ (E_g_ = 3.35 eV) is by far lower than that reported for insulators (>5 eV) [1,2,3]. The low E_g_ value of γ-Al_2_O_3_ might be due to surface defects concentration in the γ-Al_2_O_3_ particles. This can be correlated with the increase in the hybridization of the SP^3^ orbitals of γ-Al_2_O_3_ [4]. Compared with the E_g_ value of γ-Al_2_O_3_, the E_g_ of Al-Nd-5 binary oxide increases to 3.54 eV probably due to the presence of highly dispersed Nd_2_O_3_ nanoparticles, whereas the E_g_ of the binary oxides decreases to a minimum value (E_g_ = 3.13 eV) at higher Nd_2_O_3_ concentration, as observed in the Al-Nd-15 sample with E_g_ = 3.13 eV. The band gap energies of the oxides with Nd_2_O_3_ concentration between 10 and 25 wt.% were between 3.13 and 3.20 eV, and these are typical of semiconducting materials absorbing in the UV region. The calculated band gap energy value for pristine Nd_2_O_3_ (E_g_ = 3.8 eV) corresponds to electronic transitions between the valence and conduction bands [30,31]. The lower E_g_ values of the binary oxides suggest the use as photocatalysts. In particular, the low E_g_ values of binary oxides with Nd_2_O_3_ concentrations between 10 and 25 wt.% could be explained by the overlap of electronic bands from γ-Al_2_O_3_ and Nd_2_O_3_. This is suggested because the absorption bands of “4f^3^” transitions in the UV-Vis spectra of pristine Nd_2_O_3_ are shifted to lower wavelengths for approximately 13–20 nm in the UV-Vis spectra of Al-Nd-*x* binary oxides (Figure 6), indicating that Nd_2_O_3_ species are chemically interacting with γ-Al_2_O_3_. Such interaction between γ-Al_2_O_3_ and Nd_2_O_3_ might be facilitated by the sol-gel preparation method since the compounds are well-mixed and thus promote an easier interaction between species.

### 3.4. Photocatalytic Activity of Al_2_O_3_-Nd_2_O_3_ Binary Oxides in the Degradation of Diclofenac

The photocatalytic activity of γ-Al_2_O_3_, Nd_2_O_3_ and the binary oxides was evaluated in the degradation of diclofenac solutions with concentrations of 80 ppm. Previous to the photocatalytic experiments, the diclofenac solution was submitted to UV light irradiation without the use of any photocatalyst (Figure 8). 

The UV-vis spectrum indicated the stability of the drug against UV radiation; only the intensity of the band around 275 nm was slightly reduced. This band corresponds to electronic transitions of substituent groups in the diclofenac molecules, while the aromatic moiety of diclofenac is retained in the absence of a photocatalyst. The UV-Vis spectra of diclofenac after irradiation with UV light using γ-Al_2_O_3_ during a period of 80 min show that the intensity of the absorption band located at 275 nm decreases with the irradiation time and completely disappears after 60 min of UV light irradiation (Figure 9A), indicating the total degradation of diclofenac molecules after this time. However, with the presence of in neodymium in binary oxide, the absorption band located at 275 nm decreases more quickly with the photoreaction time and completely disappears after only 40 min of UV light irradiation, indicating a higher photodegradation of diclofenac (Figure 9B).

As shown in the upper plot of Figure 10, the adsorption percentage of diclofenac over Al-Nd-15 binary oxide using dark light after 80 min of exposure was less than 12%. Figure 10 also shows the relative concentration (C/C_0_) of diclofenac as a function of the photoreaction time after irradiation with UV light using γ-Al_2_O_3_, Al-Nd-*x* and Nd_2_O_3_ photocatalysts. γ-Al_2_O_3_ and Al-Nd-x binary oxides reach more than 92.0% of photoconverted diclofenac after 40 min of UV exposure. However, the relative concentration of diclofenac decreases more quickly with the Nd_2_O_3_ concentration to reach the higher drop in the Al-Nd-15 binary oxide. The photoconversion of diclofenac using γ-Al_2_O_3_ was 63.0% at 10 min of UV light exposure, whereas Al-Nd-15 binary oxide reached 82.0% at this photoreaction time. The photoconversion percentage of diclofenac at 10 min of photoreaction using Al-Nd-10 and Al-Nd-25 binary oxides was also improved with respect to that of γ-Al_2_O_3_, reaching diclofenac photoconversions of 69.0% and 78.0%, respectively. This means that the photocatalytic activity in the degradation of diclofenac using Al-Nd-*x* binary oxides with Nd_2_O_3_ contents in the range 10%–25% was improved with respect to that of γ-Al_2_O_3_ at short photoreaction times. The photoconversion percentage using Nd_2_O_3_ as a reference photocatalyst was lower than those of γ-Al_2_O_3_ and Al-Nd-x binary oxides, reaching 55.0% of photodegraded diclofenac after 10 min of UV light irradiation. 

#### Kinetic Behavior of Al_2_O_3_-Nd_2_O_3_ Binary Oxides in the Photodegradation of Diclofenac 

The kinetic data from the photodegradation of diclofenac were plotted as Ln (C_0_/C) versus reaction time (t) (Figure 11). The linear plots obtained with γ-Al_2_O_3_, Nd_2_O_3_, and the binary oxide catalysts are evidence of the pseudo first-order kinetics of the Langmuir-Hinshelwood type. The kinetics are constant with the photodegradation of diclofenac over γ-Al_2_O_3_ was 7.9 × 10^−1^ min^−1^, while the half-life time (τ_1/2_) was 8.74 min (Table 2). In the binary oxides, the reaction rate constant (k) increases with Nd_2_O_3_ concentration to reach a maximum reaction rate constant (k = 9.5 × 10^−1^) after 40 min of UV light irradiation and the half-life time decreases to a minimum (τ_1/2_ = 7.32 min). At higher Nd_2_O_3_ concentration the reaction rate constant decreases to reach a k = 4.8 × 10^−1^ min^−1^ in the pristine Nd_2_O_3_ photocatalyst. 

The kinetic studies confirm that Al_2_O_3_-Nd_2_O_3_ with Nd_2_O_3_ concentrations between 10 and 25 wt.% of Nd_2_O_3_ presented the highest activity to degrade diclofenac since these compounds achieved higher reaction rate constants and lower half-life times (Table 2). The highest activity for the photoconversion of diclofenac with 10% and 25% of Nd_2_O_3_ in the binary oxides correlates with their lower band gap energies. 

### 3.5. Recombination Rate of Electron–Hole Pairs in the Binary Oxides

(PL) spectroscopy was used to evaluate the effect of the Nd_2_O_3_ content in the binary oxides on the recombination rate of electron–hole. Figure 12 shows the PL spectra of the binary oxides and the γ-Al_2_O_3_ and Nd_2_O_3_ references. The intensity of the PL spectra of Al-Nd-5 increased with respect to that of γ-Al_2_O_3_, indicating a higher recombination rate which is close to the values of the reference Nd_2_O_3_. The intensity produced by the Al-Nd-10 sample maintained that of the γ-Al_2_O_3_ indicating lack of enhancement in the recombination rate. However, at higher Nd_2_O_3_ concentration (≥15 wt.%) the intensity decreased when the content of Nd_2_O_3_ increased suggesting that Nd_2_O_3_ acts as a separator of electron-hole pairs and favors the improvement in the photocatalytic activity.

## 4. Conclusions

The photocatalytic activity of the Al_2_O_3_-Nd_2_O_3_ binary oxides was evaluated and compared with isolated Al_2_O_3_ and Nd_2_O_3_ references. The relative concentration of diclofenac catalyzed with γ-Al_2_O_3_ and Al-Nd-x binary oxides decreased with the reaction time, reaching 92.0% of degradation after 30 min of reaction. However, the relative concentration of diclofenac decreased faster when catalyzed with Al-Nd-*x* binary oxides for Nd_2_O_3_ contents between 10 and 25 wt.%, reaching diclofenac photoconversions between 88.0% and 94.0% after only 10 min of photoreaction. Additionally, the kinetic studies confirm that the highest activity was reached with Al-Nd-*x* binary oxides with Nd_2_O_3_ concentrations of 15 and 25 wt.%. The highest activity is correlated with the lower band gap energies found in these materials, resulting from the defects concentration within the Al_2_O_3_ matrix. These low energies, along with the low recombination rates of electron-hole pairs enhanced by the Nd_2_O_3_ crystalline domains, provide evidence that the Al_2_O_3_-Nd_2_O_3_ binary oxides are suitable photocatalysts for degradation of organic compounds, especially diclofenac which is a persistent pollutant.

## Figures and Tables

**Figure 1 materials-13-01345-f001:**
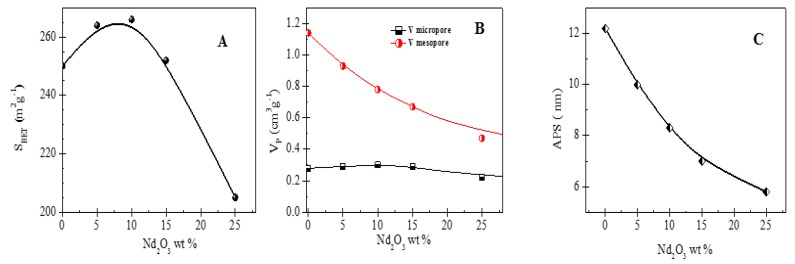
(**A**) BET specific surface area, (**B**) specific micropore and mesopore volume, and (**C**) average pore size of the Al-Nd-*x* binary oxides.

**Figure 2 materials-13-01345-f002:**
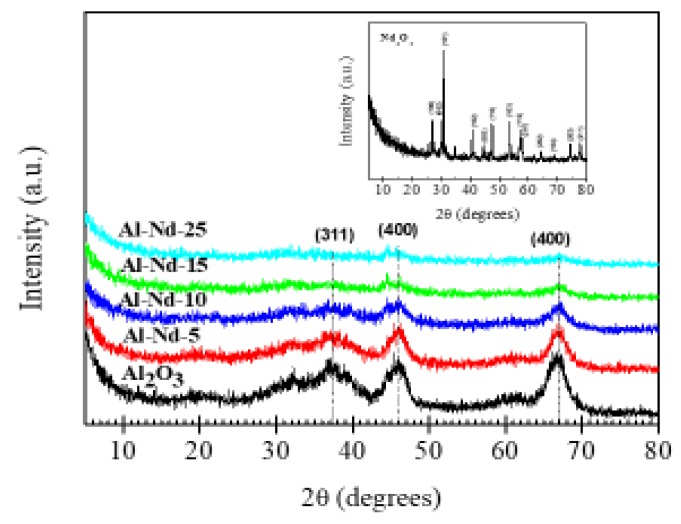
XRD patterns of γ-Al_2_O_3_ and Al-Nd-*x* binary oxides prepared by the sol-gel method.

**Figure 3 materials-13-01345-f003:**
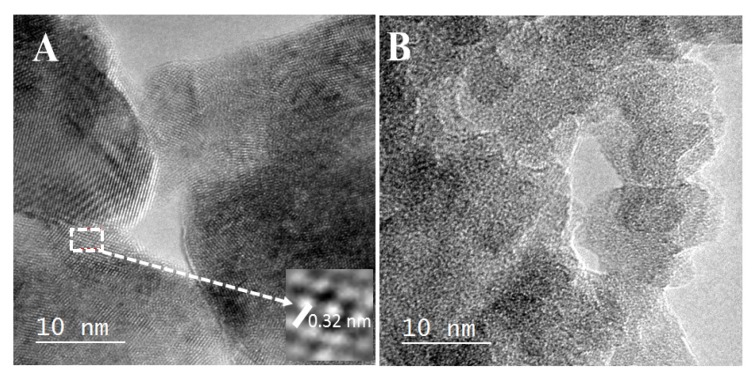
High-resolution transmission electron microscopy (HRTEM) micrographs of (**A**) Nd_2_O_3_ and (**B**) Al-Nd-15 binary oxide.

**Figure 4 materials-13-01345-f004:**
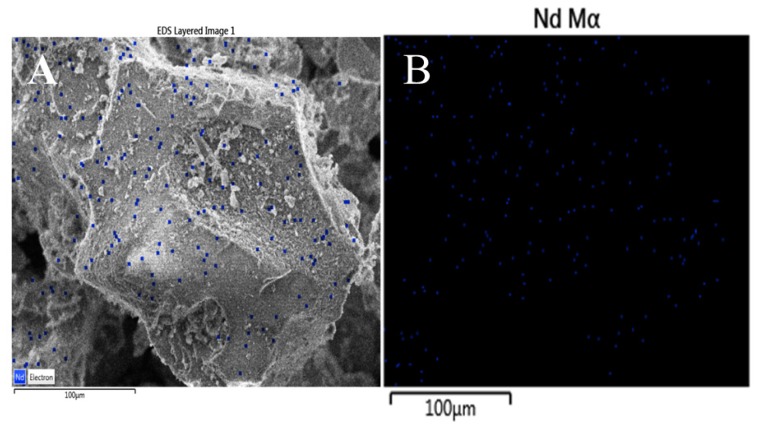
(**A**) EDS image of Al-Nd-15 binary oxide; (**B**) SEM mapping image of the element Nd Mα of the Al-Nd-15 binary oxide.

**Figure 5 materials-13-01345-f005:**
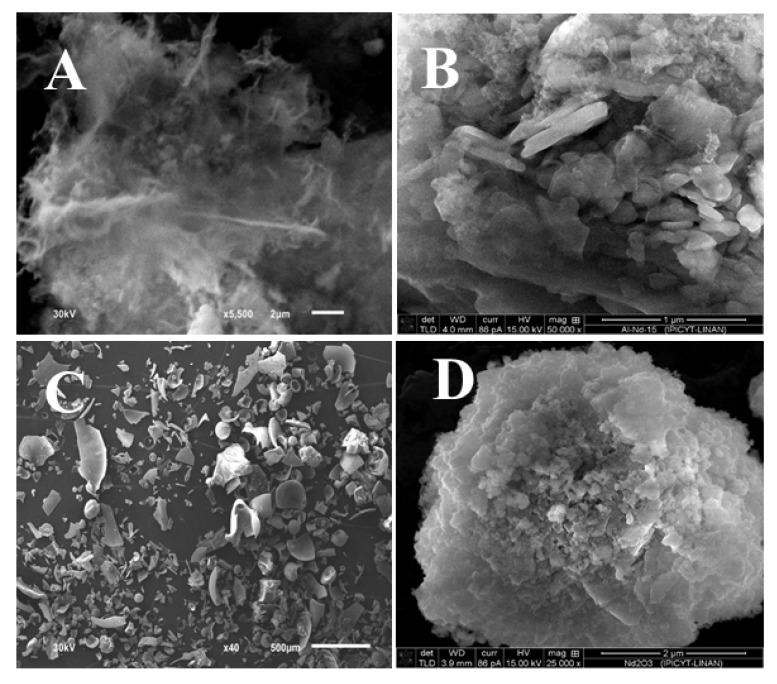
SEM micrographs of (**A**) γ-Al_2_O_3_; (**B**) Al-Nd-15; (**C**) Al-Nd-25; (**D**) Nd_2_O_3_.

**Figure 6 materials-13-01345-f006:**
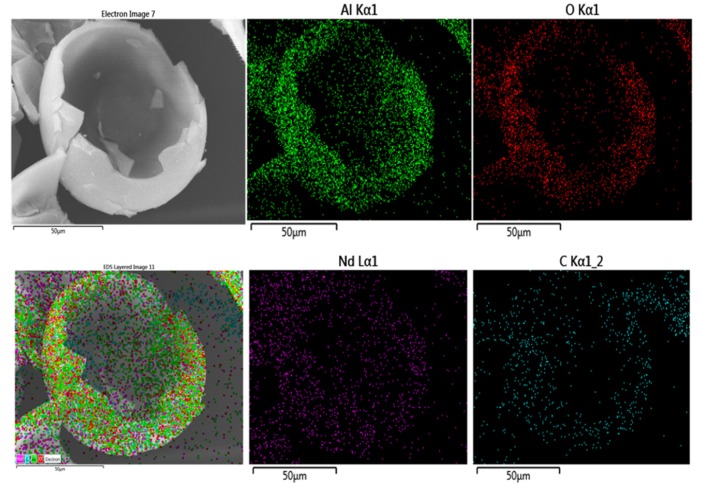
SEM mapping image by element of the Al-Nd-25 binary oxide.

**Figure 7 materials-13-01345-f007:**
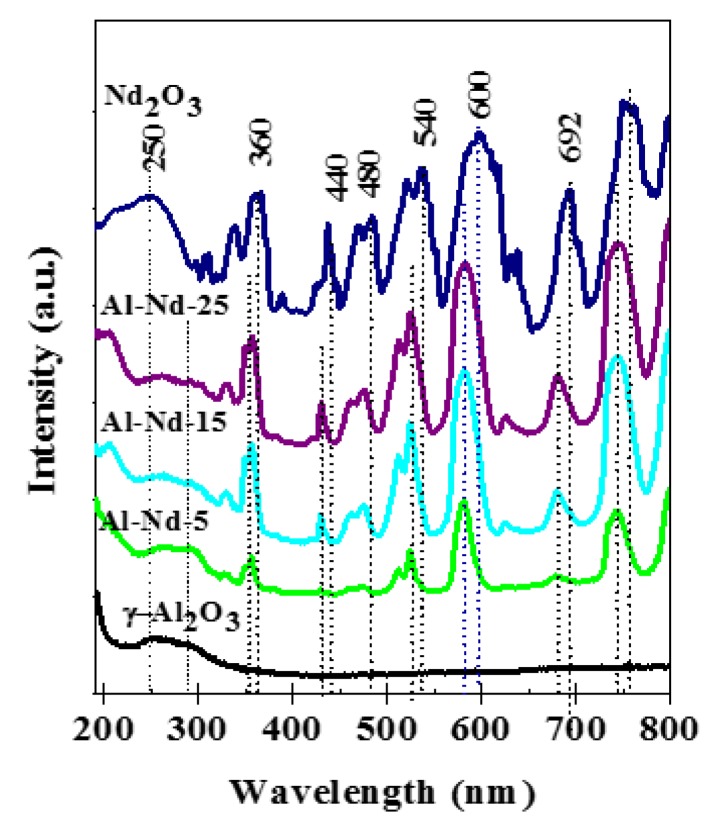
UV-Vis spectra of γ-Al_2_O_3,_ Al-Nd-*x* binary oxides and Nd_2_O_3_.

**Figure 8 materials-13-01345-f008:**
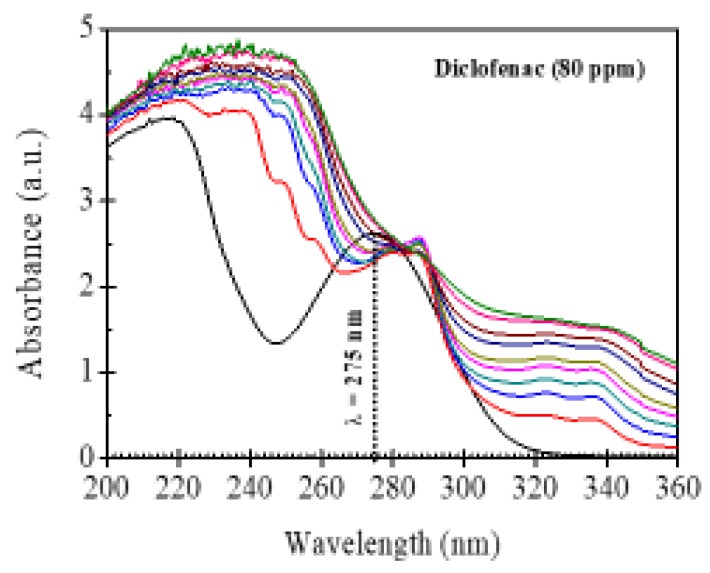
Photolysis of diclofenac using UV light irradiation without any photocatalyst.

**Figure 9 materials-13-01345-f009:**
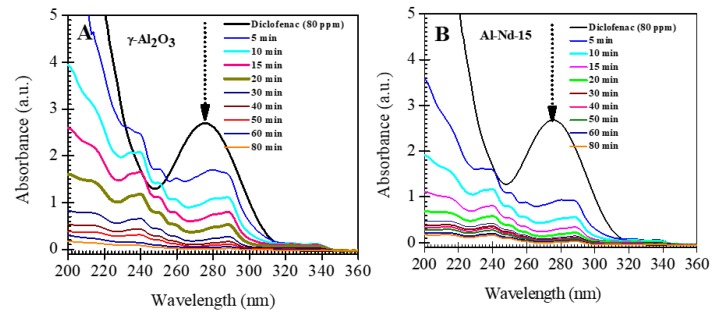
UV-Vis spectra of diclofenac after irradiation with UV light irradiation using: (**A**) γ-Al_2_O_3_; (**B**) Al-Nd-15 binary oxide.

**Figure 10 materials-13-01345-f010:**
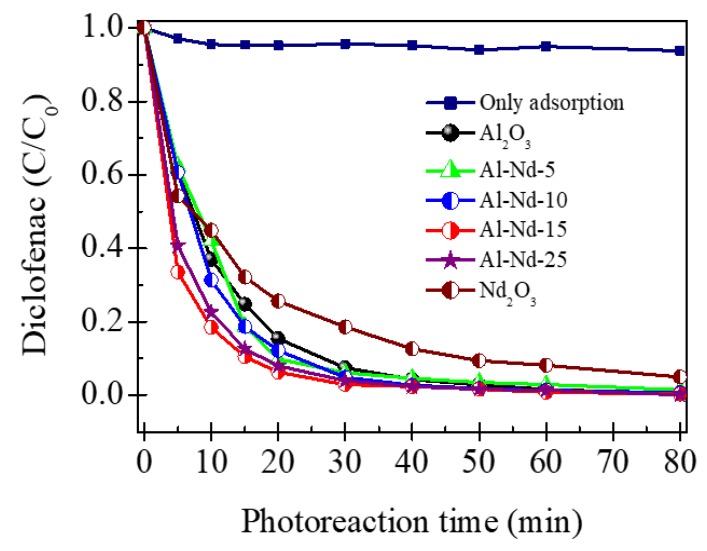
Relative concentration (C/C_0_) of diclofenac as a function of the photoreaction time using γ-Al_2_O_3_, Al-Nd-*x* binary oxides, and Nd_2_O_3_ as photocatalysts; inset plot represents the diclofenac adsorption with dark light over Al-Nd-15 binary oxide.

**Figure 11 materials-13-01345-f011:**
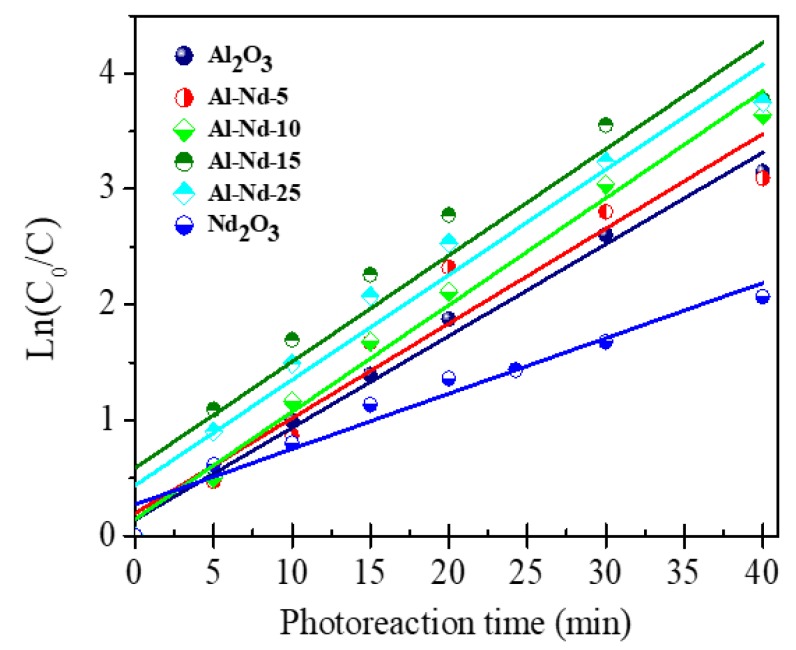
Ln (C_0_/C) vs. photoreaction time (t) after photodegradation of diclofenac using γ-Al_2_O_3_, Al-Nd-*x* binary oxide and Nd_2_O_3_.

**Figure 12 materials-13-01345-f012:**
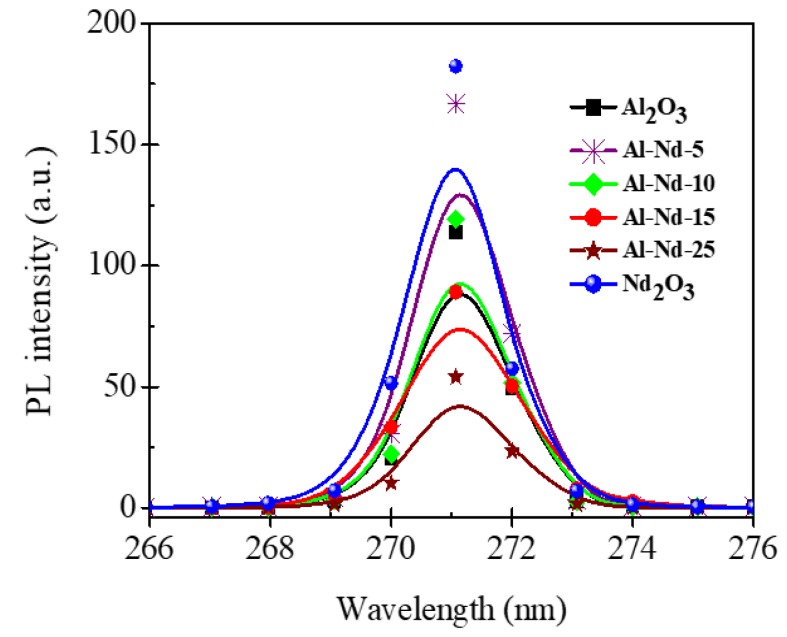
PL spectra of γ-Al_2_O_3_, Al-Nd-*x* binary oxides and Nd_2_O_3_.

**Table 1 materials-13-01345-t001:** Band gap energy of Al_2_O_3_-Nd_2_O_3_ binary oxides prepared by the sol-gel method.

Material	Band Gap Energy (eV)
γ-Al_2_O_3_	3.35
Al-Nd-5	3.54
Al-Nd-10	3.20
Al-Nd-15	3.13
Al-Nd-25	3.19
Nd_2_O_3_	3.98

**Table 2 materials-13-01345-t002:** Rate constant (k) and half-life time (τ_1/2_) in the diclofenac photo-degradation reaction using Al_2_O_3_-Nd_2_O_3_ binary oxides.

Photocatalyst	k × 10^−2^ (min^−1^)	τ_1/2_ (min)
Al_2_O_3_	7.9	8.74
Al-Nd-5	8.2	8.46
Al-Nd-10	9.3	7.50
Al-Nd-15	9.5	7.29
Al-Nd-25	9.1	7.61
Nd_2_O_3_	4.8	14.50

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
