# Peer review of "Photocatalytic Degradation of Diclofenac Using Al2O3-Nd2O3 Binary Oxides Prepared by the Sol-Gel Method"

_materials, 2020, doi:10.3390/ma13061345_

Round 1

Reviewer 1 Report

Title: Photocatalytic degradation of diclofenac using Al2O3-2 Nd2O3 binary oxides prepared by the sol-gel method

 Brief summary:

This work reports the synthesis of a Al2O3-Nd2O3 by sol-gel method, its complete characterization and a study of its photocatalytic activity in the degradation of diclofenac by UV. The addition of Nd2O3 improved the photocatalytic activity, due to the lowering of the band gap, being Nd2O3 a separator of charges which decreases the recombination rate.

General comment:

The characterization of the material is complete and well reported. My biggest concern is about the photocatalytic result, because there is an increase of degradation of only 2 %. Moreover, my most important doubt is if the photolysis has been considered, and, also, which is the result obtained in the dark (i.e. the absorption of the diclofenac by the material).

If done, report these results, giving information about the degradation pathway using the photocatalyst. 

Most important lacks:

An increase of the diclofenac degradation of 2% is very low.

Results reported in the abstract appear approximate.

which is the contribution of photolysis (UV only)? if considered, which are the by-products of diclofenac degradation obtained with this photocatalytic system?

For all these reason I think that this manuscript needs to be revised (major revision) before publication.

In the following lines, other notes, comments and questions.

Abstract:

About the band gap: say directly how much decreased (from and to which value), and which is the relation between decrease and Nd2O3 concentration.

An increase of 2 % of degradation is very low to justify the real potential of the material.

Avoid the use of comparatives but give immediately numbers/amounts: higher reaction rate constants, than what?

The explanation of the results appears rather approximate in the abstract.

Conclusions:

Conclusions repeat what is reported in the abstract.

Al-Nd-x binary oxides showed the highest activity, does it mean that this is the best system for diclofenac degradation?

Materials and methods:

  • Synthesis: it is always better to provide precise quantities in order to make the experiments
  • 200 mL of the diclofenac dissolution: which is the concentration?
  • To follow the photodegradation of diclofenac as a function of time, samples of the irradiated diclofenac solution were taken each 5 min from the photo-reactor using a filter for the first 20 min, after that the samples of irradiated diclofenac solution were taken each 10 min for a period of 60 min: this sentence is unclear, reformulate.

Results:

  • Samples with 10, 15 and 25 % of Nd are different but they are presented together talking about the photocatalytic activity. This is not very clear.
  • Figure 5: the scale is not clear. Are the images at the same magnification?
  • This absorption band is attributed to electronic charge transfer or the presence of a high density of defects in the structure of γ-Al2O3 ? can be both? Why?
  • Why an addition of Nd 5 % causes an increasing of the band gap?
  • Which is the contribution of photolysis (UV only) in case of ciclofenac
  • Are the by-products considered?

Author Response

Reviewer 1

The characterization of the material is complete and well reported. My biggest concern is about the photocatalytic result, because there is an increase of degradation of only 2 %. Moreover, my most important doubt is if the photolysis has been considered, and, also, which is the result obtained in the dark (i.e. the absorption of the diclofenac by the material).

Answer:

(1). A plot of photolysis of diclofenac using UV light irradiation without any photocatalyst has been included in the manuscript. The UV-vis spectrum indicated the stability of the drug against UV radiation; only the intensity of the band around 275 nm was slightly reduced; this band corresponds to electronic transitions of substituent groups in the diclofenac molecules, while the aromatic moiety of diclofenac is retained in the absence of a photocatalyst.

(2). A representative plot of diclofenac adsorption using dark light has been included in the plot of relative C/C0 vs. photoreaction time. As shown in the upper plot of Fig. 10, the adsorption percentage of diclofenac over Al-Nd-x materials using dark light after 80 min of exposure was less than 12 %.

(3). The discussion concerning the photoconversion percentage of diclofenac has been corrected in the manuscript, the photoconversion percentage of diclofenac over Al-Nd-x with Nd2O3 concentration in the range 10 – 25 wt %  was improved with respect to that of g-Al2O3 at short photoreaction times as follows:

Figure 10 also shows the relative concentration (C/C0) of diclofenac as a function of the photoreaction time after irradiation with UV light using g-Al2O3, Al-Nd-x and Nd2O3 photocatalysts. g-Al2O3 and Al-Nd-x binary oxides reaches more than 92.0 % of photoconverted diclofenac after 40 min of UV exposure. However, the relative concentration of diclofenac decreases more quickly with the Nd2O3 concentration to reach the higher drop in the Al-Nd-15 binary oxide. The photoconversion of diclofenac using g-Al2O3 was 63.0 % at 10 min of UV light exposure, whereas Al-Nd-15 binary oxide reached 82.0 % at this photoreaction time. The photoconversion percentage of diclofenac at 10 min of photoreaction using Al-Nd-10 and Al-Nd-25 binary oxides was also improved with respect to that of g-Al2O3, reaching diclofenac photoconversions of 69.0 % and 78.0 % respectively. It means that the photocatalytic activity in the degradation of diclofenac using Al-Nd-x binary oxides with Nd2O3 contents in the range 10 – 25 % was improved with respect to that of g-Al2O3 at short photoreaction times. The photoconversion percentage using Nd2O3 as reference photocatalyst was lower than those of g-Al2O3 and Al-Nd-x binary oxides reaching 55.0 % of photodegraded diclofenac after 10 min of UV light irradiation.

If done, report these results, giving information about the degradation pathway using the photocatalyst. 

Answer:

Photolysis result has been included as the Figure 8 of the manuscript and a representative plot of diclofenac adsorption with dark light using Al-Nd-15 binary oxide was also included as inset in Fig. 10.

Most important lacks:

An increase of the diclofenac degradation of 2% is very low.

Results reported in the abstract appear approximate.

Answer:

The interpretation of photoconversion data of diclofenac and its discussion was corrected in the manuscript; the abstract and the conclusions were also corrected. The photoconversion percentage of diclofenac over Al-Nd-x with Nd2O3 concentration in the range 10 – 25 wt %  was improved with respect to that of g-Al2O3 at short photoreaction times. The diclofenac photoconversion percentage over g-Al2O3 after 10 min of reaction was improved in 15 – 20 % when Al-Nd-15 and Al-Nd-25 were used.

Which is the contribution of photolysis (UV only)? if considered, which are the by-products of diclofenac degradation obtained with this photocatalytic system?

Answer:

The UV-vis spectrum indicated the stability of diclofenac molecule against UV light radiation; only the intensity of the band around 275 nm was slightly reduced; this band corresponds to electronic transitions of substituent groups in the diclofenac molecules, while the aromatic moiety of diclofenac molecule is retained in the absence of a photocatalyst. At present, we could not to identify the intermediates and products of diclofenac photodegradation due to the lack HPLC equipment, however we are working in this way in order to identify the possible reaction intermediates.

In the following lines, other notes, comments and questions.

Abstract:

About the band gap: say directly how much decreased (from and to which value), and which is the relation between decrease and Nd2O3 concentration.

Answer:

The Abstract was corrected concerning to the band gap energies as follows: The inclusion of Nd2O3 in the aluminum oxide matrix in the 10 – 25 % range reduced the band gap energies from 3.38 eV for the g-Al2O3 to values as low as 3.13 – 3.20 eV.

An increase of 2 % of degradation is very low to justify the real potential of the material.

Answer:

Concerning to the interpretation of photodegraded diclofenac percentage by g-Al2O3 and Al-Nd-x binary oxides, the abstract and the conclusions were corrected. The photoconversion percentage of diclofenac over Al-Nd-x with Nd2O3 concentration in the range 10 – 25 wt %  was improved with respect to that of g-Al2O3 at short photoreaction times. Although all the materials photoconvert more tan 92 % of diclofenac after 30 min of photoreaction, however, the diclofenac photoconversion percentage was improved in 15 – 20 % with respect to that of g-Al2O3 after 10 min of reaction when Al-Nd-15 and Al-Nd-25 were used.

 Avoid the use of comparatives but give immediately numbers/amounts: higher reaction rate constants, than what?

The explanation of the results appears rather approximate in the abstract

The abstract has been corrected in the manuscript, I hope the present form of the manuscript improves the explanation of the results in the abstract.

Thank you very much in advance for your kind attention for the revision of this manuscript, I appreciate your observations and suggestions.

Reviewer 2 Report

The research work in this paper is very well presented and explained. The subject that was studied is interesting and significant for the environment protection. The methods used are adequate and results are well explained and discussed.  

There is a few minor disadvantages that should be added in the paper.

In the section 2 Materials and Methods, the list of used materials/ chemical is missing. It is important to know the main properties and grade of quality of used materials.

In Fig. 2 the spectra of bare Nd2O3 is too small should be slightly increased.  

Then, discussion related to dispersion of Nd2O3 in Al2O3, Fig. 4 and 5. are poorly discussed. Meaning, for the images Fig 4 it is said that dispersion is very good and on the Fig 5 it is said that agglomeration occurred. This should clarify why uniform dispersion occurs, and on the other hand, why particle aggregation occurs? What does the literature say about it?

Author Response

Reviewer 2

In the section 2 Materials and Methods, the list of used materials/chemical is missing. It is important to know the main properties and grade of quality of used materials.

Answer:

The grade of quality of reagents and solvents has been included in the Experimental section of the manuscript.

In Fig. 2 the spectra of bare Nd2O3 is too small should be slightly increased.

Answer:

The spectra of bare Nd2O3 in the inset of Fig. 2 has been increased.

Then, discussion related to dispersion of Nd2O3 in Al2O3, Fig. 4 and 5. are poorly discussed. Meaning, for the images Fig 4 it is said that dispersion is very correction made why uniform dispersion occurs, and on the other hand, why particle aggregation occurs? What does the literature say about it?

Answer:

The discussion regarding to dispersión of Nd2O3 in Al2O3 and the particle aggregation has been improved, a SEM mapping image by element of the Al-Nd-x binary oxide with high concentration of Nd2O3 has been included in the manuscript in order to clarify the particle aggregation.

Thank you very much in advance for your kind attention for the revision of this manuscript, I appreciate your observations and suggestions.

Round 2

Reviewer 1 Report

Dear authors,

as I wrote to the editor,

I think that you improved sufficiently the paper. The biggest lack about the photolysis and dark adsorption are explained.

Data are discussed more in deep.

I'm not completely sure about the real strength of the paper from a novelty point of view, but now it is complete and useful to consult.

From here, the study of the by-products will be very interesting.

with my best regards